# The Relationship of Temperament and Character, Parental Stress, and Mental Health Problems with Attachment Disorders among Children

**DOI:** 10.3390/ijerph192315458

**Published:** 2022-11-22

**Authors:** Martin Schröder, Süheyla Seker, Delfine d’Huart, Yonca Izat, Margarete Bolten, Klaus Schmeck, Marc Schmid

**Affiliations:** 1Department of Child and Adolescent Psychiatry Research, University Psychiatric Clinics Basel, University of Basel, 4002 Basel, Switzerland; 2Vivantes Clinic Friedrichshain, Child and Adolescent Psychiatry Berlin, 10249 Berlin, Germany

**Keywords:** attachment, personality, temperament, character, developmental psychology, foster care, mental health problems, parental stress

## Abstract

According to Cloninger’s model, personality is conceptualized in temperament and character traits contributing to a child’s psychosocial development. Additionally, parent–child interaction is important for the child’s socio-emotional development. To date, the relationship between attachment and temperament and character for child mental health development and its effects on parents remains mostly unclear. The aim of the present study was thus to examine the relationship of attachment, temperament and character, parental stress, and mental health problems among 125 children (mean age = 7.14 years) in Switzerland. Temperament and character, attachment disorder (symptoms), parental stress, and mental health problems were assessed with psychometric questionnaires; attachment was assessed with an additional observational measure. Descriptive characters of the sample were presented, and group differences and correlations were computed. For temperament traits, results revealed significant group differences for novelty seeking and persistence and attachment disorder types. For character traits, the findings showed significant group differences for self-directedness and cooperativeness and attachment disorder types. Moderate effect sizes for groups differences were found. Further, the mixed-type (inhibited and disinhibited) and inhibited attachment disorder type were the most burdened groups. The present findings suggest that temperament and character traits, as well as parental stress and mental health problems are associated with the occurrence of attachment disorders among children. Future longitudinal studies with larger samples are needed to examine the causal relationships of temperament and character with attachment, including person-related and environmental factors among children.

## 1. Introduction

In personality psychology, many different but also overlapping attempts at defining human personality have developed since its origins due to different substantive emphases. The most relevant paradigms here are the psychoanalytic, behavioral/social learning, biological, evolutionary, neuroscientific and cognitive approaches. Consequently, there cannot be one definition of personality, as it varies according to the paradigm. However, a simple description of personality that summarizes these different paradigms makes it clear that personality is characterized by consistent patterns of behavior across time, space and situations that develop from the interaction between environment and individual in intrapersonal emotional, motivational and cognitive processes [1,2].

Within personality theory, Robert C. Cloninger’s psychobiological model of personality combines many of the paradigms listed above. Cloninger’s concept of personality [3] makes the helpful conceptually operationalized distinction between temperament traits (predominantly genetic influences) and character traits (predominantly environmental influences) and considers the mutual influences of these traits in the process of personality development. Temperament consists of relatively consistent, basic and individual dispositions that underlie and modulate the expression of activity, reactivity, emotionality and sociability. Essential elements of temperament are present early in life, and these elements are strongly influenced by neurobiological factors. In Cloninger’s model, temperament traits include novelty seeking, harm avoidance, reward dependence and persistence. Novelty seeking represents the level of behavioral activation, harm avoidance represents the behavioral inhibition system, reward dependence represents the maintenance of behavior through social reinforcement, and the maintenance of behavior without external influences, i.e., through intrinsic motivation, is called persistence [4]. In a recent publication, Cloninger et al. [5] define temperament as the disposition of a person to learn how to behave, react emotionally, and form attachments automatically by associative conditioning.

Character, the second basic personality dimension, includes self-concepts and individual differences in goals and values that influence decision-making, intentions and the meaning of what is experienced in life. Differences in character are said to be influenced by socio-cultural learning and to mature continuously in the life process. In Cloninger’s model of personality, character traits include self-directedness, cooperativeness, and self-transcendence. Self- directedness describes responsible and mature behavior and self-acceptance. The engagement with the social environment is depicted in cooperativeness, which is defined as helpful, tolerant and empathetic behavior. Self-transcendence, indicates the awareness of spiritual values, i.e., the extent of an individual’s ability to recognize that other universal and transcendent values exist alongside the self and the social environment [4]. Impairments in the two character dimensions of self-directedness and cooperativeness are strong indicators of the presence of a personality disorder [3] which are closely related to attachment styles [6]. Cloninger’s concept of personality postulates the interrelationship of continuous transactions between these four temperaments and character traits and three character traits in the process of personality development maturation [3].

There is still an unresolved question in developmental psychology about the relationship between attachment and personality or temperament and its developmental effects later in life [7,8]. On the one hand, attachment has been described as an expression of temperament development [9]. Whereas, on the other hand, temperament development has been found to have little influence on attachment [10]. As a result, studies so far revealed contradictory findings, leading to ambiguous conclusions regarding the relationship between temperament, character and attachment [7,8]. As a possible explanation for these unclear results, Groh et al. [7] discussed the different operationalization of temperament. In addition, Bowlby [11] highlighted the complexity of parent–child interaction from an attachment perspective. Children have genetic dispositions that significantly influence their interaction with the environment, while biographical experiences of interaction with their parents substantially shape children’s thoughts, feelings, and behaviors in close relationships. Concerning the relationship of attachment style and character, Chotai et al. [12] describe that secure attachment is positively correlated with cooperativeness and self-transcendence. Anxious/ambivalent attachment was correlated negatively with self-directedness.

This background emphasizes the close conceptual link between attachment theory and personality theory [13,14,15]. Accordingly, a secure attachment representation may be considered as a protective resilience factor. In contrast, disorganized attachment or even an attachment disorder could be perceived as a substantial risk factor for personality development and mental health. Secure attachments can, thus, compensate for children with challenging temperament dimensions. Conversely, highly insecure attachments can negatively influence children with easy temperament, which is consequently reflected in the development of the character dimension. To date, however, it is unclear whether genetic personality traits (such as temperament) are either independent predictors of interpersonal interactions, potential moderators of the relationship between attachment and interpersonal behavior, or rather a combination of these processes.

From the perspective of attachment theory, considerable importance is given to the aspects of sensitivity and responsiveness of parents towards their children, which have an impact on the child’s mental health. The less parents feel subjectively stressed in their parenting behavior, the more responsively, sensitively and with a larger emotional range of action they can respond to the child’s emotional needs. Parental stress not only has far-reaching negative consequences for the well-being of the individuals themselves, but also negatively affects a child’s psychosocial development and physical health. This, in turn, increases the parental burden, which, eventually, may result in a vicious circle [16]. An increasing body of literature indicates that children placed in child welfare, as well as in child and adolescent psychiatry, have experienced high psychosocial burden within their parental homes. Due to the accumulation of psychosocial burdens for children within the child welfare system or in psychiatric care systems, this makes them a particularly vulnerable group compared to the general child population for the development of attachment disorders, personality disorders, and mental health problems [17,18,19,20,21].

To sum up, a vast body of studies has shown that temperament and character as well as parent–child interaction are crucial to children’s psychosocial development. However, to the best of our knowledge, the relationship of temperament and character traits, parental stress, and child mental health problems with and attachment disorders, in particular, remains mostly unclear. Against the background of previous literature and Cloninger’s theory, the aim of this study was twofold. To pursue this aim, we posed the following two research questions:(a).What is the relationship between attachment disorder and representations with temperament and character traits, parental stress, and mental health problems among children?(b).How do children with attachment disorder types differ regarding their temperament and character traits, parental stress, and mental health problems?

## 2. Materials and Methods

### 2.1. Study Design

Data for the present study originated from the “Validation of the Relationship Problems Questionnaire in various high-risk populations” study which was conducted between 2013 and 2015 at the Department of Child and Adolescent Psychiatry Research of the University Psychiatric Clinics, University of Basel, Switzerland.

The aim of the study was to assess attachment representations and disorders among a total of 152 children from the general population (control sample; *n* = 34, mean age = 7.52 years), foster care (foster care sample; *n* = 32, mean age = 7.52 years), and clinical settings (clinical sample; *n* = 86, mean age = 7.24 years). Participants were recruited in the German-speaking part of Switzerland and the city-state Berlin, Germany. Participants from the general population were recruited by announcements in day-care centers, playgroups, kindergartens, primary schools, and sport clubs. Children in the foster care sample were recruited by different stakeholders in foster care using newsletters and references to highly frequented websites. Finally, participants from clinical settings were recruited in in- and outpatient units.

The study design included a combination of questionnaires and face-to-face assessments. The children only participated in the face-to-face assessment and did not fill out any questionnaires. The child explanation of the procedure, verbally obtaining the child’s consent for implementation, introduction, implementation, and completion of the procedure required an average of 60 min and was fully videotaped.

In contrast, the children’s caregivers participated in the structured interview for axis-I disorders in parallel and received the necessary questionnaires in advance for processing. The structured interview with the K-DIPS was directly written down in the corresponding coding system and the questionnaires were computerized for analysis.

The research staff was trained and certified in the assessment procedures to ensure high quality data. Children and their caregivers were thoroughly informed about the study, and the caregivers of the children gave their written informed consent. Explanations and decisions regarding participation of the children were videotaped and transcribed. As compensation for their participation, participants took part in a raffle of zoo vouchers. The Ethics Committee Northwest and Central Switzerland (German: *Ethikkommission Nordwest- und Zentralschweiz* [EKNZ]) approved the study before commencement (EKNZ reference number: 53/12).

### 2.2. Participants

The final sample of the present study consisted of a total of 125 children (age range = 4–10 years) including children from the general population (*n* = 30), foster care (*n* = 32), and child and adolescent psychiatric settings (*n* = 63). Predefined exclusion criteria were insufficient knowledge of the German language, intelligence quotient ≤ 70, and autism spectrum disorders.

Due to missing data, some participants were excluded from the current analyses. A total of 12.5% missing data for our study variables were found (rates ranging from 3.2–16.0% for individual variables). We compared included participants with excluded participants using χ^2^-square and independent sample *t*-tests, respectively. The attrition bias analysis showed that among included participants, the proportion of females was significantly higher compared to those excluded from the study (see Appendix A). For age, nationality, mental health problems, personality traits, attachment disorder symptoms, attachment disorder, attachment disorder type, attachment representation, and parental stress, excluded participants did not differ significantly from those included in the study. The present analyses were, thus, conducted with complete cases.

### 2.3. Measures

*Attachment disorder*. Attachment disorders were assessed with the German structured interview for axis-I disorders (German: *Diagnostisches Interview psychischer Störungen im Kindes-und Jugendalter*; K-DIPS; [22]). The present study used the unpublished research version of the K-DIPS because it provides further information from the primary caregiver on specific aspects of attachment disorder, namely inhibited (A1 criterion of the Diagnostic and Statistical Manual of Mental Disorders, fourth edition, text revision [DSM-IV-TR]) or disinhibited (A2 criterion of DSM-IV-TR) subtypes. In addition, the research version provides C-criterion related information for pathogenic care in the form of continuous disregard of the emotional (C1) and physical (C2) basic needs of children, as well as repeated changes of the most important caregivers (C3) before the age of five. An attachment disorder was only assigned, if both Criterion A and at least one of the C criteria were met. Children who fulfilled the diagnostic criteria for both subtypes of attachment disorder were classified as a mixed subtype. The K-DIPS has good validity and reliability for axis-I disorders (parent version, kappa = 0.88 to 0.95; [23]).

*Attachment disorder symptoms*. Attachment disorder symptoms were assessed with the German version of the *Relationship* Problems *Questionnaire* (RPQ; [24]) which is a valid screening tool to identify children with clinical symptoms of attachment disorders [25]. The RPQ is an economical caregiver-report screening questionnaire, which assesses relationship problems and behaviors that typically constitute symptoms of an attachment disorder and enables differentiation between the disinhibited and inhibited subscales. Four possible responses can be scored on a scale ranging from 0 to 3 and the sum of the 10 items yields the RPQ total score. In the present study, we administered the short German 10-items version of the RPQ [26]. The German version of the RPQ shows high internal consistencies for the RPQ total score (Cronbach’s α = 0.81) as well as for the disinhibited (Cronbach’s α = 0.86) and inhibited (Cronbach’s α = 0.74) attachment behavior subscales [27].

*Attachment representations*. Attachment representations were assessed by the observational measure German Attachment Story Completion Task (GEV-B) for children aged five to eight years [28]. The procedure is based on the addition of stories depicting attachment-relevant situations. The child’s narratives were used to infer attachment representations (ambivalent, secure, avoidant, and disorganized). Previous studies showed good 168 inter-rater reliability with regard to their assessments of the attachment security value [28].

*Temperament and character traits*. Temperament and character traits were assessed with the German version of *the Junior* Temperament *and Character Inventory* for children aged 3–6 years and 7–11 years (JTCI 3–6, 7-11 R; [29]). The JTCI 7-11 R assesses the four temperament traits, novelty seeking, harm avoidance, reward dependence and persistence and the three character traits, self-directedness, cooperativeness and self-transcendence traits according to Cloninger’s psychobiological model of personality [6]. The questionnaire has shown good psychometric properties in a German normative and clinical sample [30].

*Parental Stress.* Similar to the original version [31], the Parental Stress Scale (PSS; [32]) is an 18-item questionnaire assessing parents’ feelings about their parenting role, exploring both positive aspects (e.g., emotional benefits, personal development) and negative aspects of parenthood (e.g., demands on resources, feelings of stress). The items are rated on a five-point Likert scale from1 = ‘not at all true’ to 5 = ‘true exactly’. A high total score across all items reflects a greater degree of stress caused by parenthood. The German version of the PSS showed a good internal consistency and construct validity [33].

*Mental health problems*. Emotional and behavioral problems were assessed with the German version of the screening instrument *Child Behavior* Checklist (CBCL/4-18; [34]). The CBCL/4-18 is part of the Achenbach System of Empirically Based Assessment (ASEBA). The items can be assigned to eight subscales (i.e., social withdrawal, somatic complaints, anxiety/depression, social problems, thought problems, attention problems, delinquent behavior, and aggressive behavior), which are summarized in two broad band scales (internalizing and externalizing) and a total score. The internal consistency and construct validity of the German version of the CBCL was found to be good in previous studies [35].

### 2.4. Statistical Analyses

First, the Shapiro–Wilk’s test showed non-normality of our data. Thus, we calculated descriptive information to examine group differences in socio-demographic characteristics, personality traits, attachment disorder symptoms, attachment disorder types, attachment representation, parental stress, and emotional and behavioral problems by sub-groups using Pearson’s χ^2^-square or the Mann–Whitney U test where appropriate.

Second, bivariate correlations between continuous variables were examined with Pearson’s correlations and between categorical variables with Spearman’s correlations.

Third, to compare group differences between personality traits and attachment disorder types, we applied a Kruskal–Wallis test due to the non-normality of the data using groups of attachment disorder types and attachment relationship types as the between-subject factor, respectively. To control for the Type-I-error, the multiple mean comparisons of significant results of the analysis of variance were tested using a post hoc analysis with the Bonferroni–Holm correction. Effect sizes for the Kruskal–Wallis tests were calculated as eta-square (η^2^) and transformed to Cohen’s *d* [36]. The magnitude of Cohen’s *d* is interpreted as following: large effect size of *d* = 0.8, medium effect size of *d* = 0.5, and small effect size of *d* = 0.2.

All analyses were calculated with the statistical software R (version 4.0.2; [37]). Statistical significance was set to *p* < 0.05 for all analyses.

## 3. Results

### 3.1. Sociodemographic and Descriptive Characteristics

Findings regarding sociodemographic and descriptive characteristics of the total sample and group differences for the sub-samples are presented in Table 1. First, the mean age of the total sample was 7.14 years, including 49 (39.2%) females and 66 children (52.8%) with Swiss nationality. The sub-samples (i.e., clinical, foster care, and control) differed significantly regarding age (χ^2^(2) = 13.92, *p* < 0.001), gender (χ^2^(2) = 11.20, *p =* 0.003), and nationality (χ^2^(2) = 19.10, *p* < 0.001).

Second, the sub-samples significantly differed regarding mental health problems (internalizing problems: χ^2^(123) = 25.25, *p <* 0.001, externalizing problems: χ^2^(2) = 26.96, *p* < 0.001, total problem behavior: χ^2^(2) = 33.10, *p* < 0.001), attachment disorder symptoms (total scale: χ^2^(2) = 24.70, *p* < 0.001, disinhibited subscale: χ^2^(2) = 24.70, *p* < 0.001, inhibited subscale: χ^2^(2) = 14.04, *p* < 0.001), and attachment disorder (χ^2^(2) = 8.50, *p* = 0.01).

### 3.2. Relationship of Temperament and Character, Parental Stress, and Mental Health Problems with Attachment Disorders

Bivariate correlations between all study variables are included in Table 2. The attachment disorder symptoms total scale correlated significantly and positively with novelty seeking (*r* = 0.53, *p* < 0.001), and negatively with cooperativeness (*r* = −0.50, *p* < 0.001) and self-directedness (*r* = −0.48, *p* < 0.001). The attachment disorder symptoms disinhibited subscale was associated significantly and positively with novelty seeking (*r* = 0.38, *p* < 0.01) and negatively with self-directedness (*r* = −0.34, *p* < 0.05). The attachment disorder symptoms inhibited subscale correlated significantly and positively with novelty seeking (*r* = 0.52, *p* < 0.001), and negatively with cooperativeness (*r* = −0.55, *p* < 0.001) and self-directedness (*r* = −0.46, *p* < 0.001). Finally, the attachment disorder type was significantly and negatively associated with cooperativeness (*r* = −0.33, *p* < 0.05).

*Group differences for temperament and character traits, mental health problems, and parental stress with attachment disorder types.*Table 3 shows the scores of temperament and character traits stratified by attachment disorder type. For the temperament traits, the results showed that significant group differences were found for novelty seeking (χ^2^(3) = 14.15, *p* = 0.002, *d* = 0.63) and persistence (χ^2^(3) = 9.46, *p* = 0.02, *d* = 0.46) and the attachment disorder types. For character traits, the findings revealed significant group differences were found between cooperativeness (χ^2^(3) = 18.00, *p* < 0.001, *d* = 0.74) and attachment disorder types. Additionally, significant group differences were found between internalizing problems (χ^2^(3) = 13.37, *p* = 0.003, *d* = 0.63), externalizing problems (χ^2^(3) = 16.04 *p* = 0.001, *d* = 0.70), total problem behavior (χ^2^(3) = 18.10, *p* < 0.001, *d* = 0.13), and parental stress (χ^2^(3) = 10.64, *p* = 0.01, *d* = 0.51) and the attachment disorder types.

The Bonferroni–Holm post hoc analysis showed that the mixed type of attachment disorder showed significantly higher scores for novelty seeking (*p* = 0.003), internalizing (*p* = 0.02), externalizing (*p* = 0.003), total problems behavior (*p* < 0.001), and parental stress (*p* = 0.01), and lower scores for persistence (*p* = 0.03), cooperativeness (*p* = 0.001) compared to the group with no attachment disorder. Furthermore, the Bonferroni–Holm post hoc analysis revealed that the inhibited attachment disorder type had significantly higher levels of externalizing problems (*p* = 0.04) and lower levels of cooperativeness (*p* < 0.04) compared to the group without an attachment disorder. Lastly, the Bonferroni–Holm post hoc analysis revealed that the mixed type of attachment disorder showed significantly higher levels of novelty seeking (*p* = 0.003), externalizing problems (*p* = 0.04), and total problem behavior (*p* < 0.03), and lower levels of cooperativeness (*p* = 0.04) and compared with the disinhibited attachment disorder type.

## 4. Discussion

The main aim of the present study was to investigate the relationship of temperament and character, parental stress, and mental health problems with attachment disorders among children.

First, our findings showed that temperament and character traits are associated with attachment disorders, suggesting that these personality traits can increase the risk for the occurrence of attachment disorders among children. According to attachment theory [11], temperament research [10] and personality theory [3,13,38], our finding is less surprising, as the development of attachment and character is significantly influenced by the surrounding environment, as well as interactions and socio-cultural learning experiences. Consequently, this connection is consistent with the meta-analyses from Groh et al. [7] and van IJzendoorn et al. [8]. This confirms the high influence of environmental factors on the development of a child’s character.

Biological aspects (i.e., temperament) contribute to an individual’s basic disposition, which interact with the environment and contribute to personality development (i.e., character dimension) [11]. The association between temperament and character traits with attachment disorder in our study also endorses the high proportion of genetic predisposition and the effects of parent–child interaction. From the perspective of the differential susceptibility theory [8], Pluess and Belsky [39] assume that children have an inherent different susceptibility to protective and risk factors from the environment (parent–child relationship). This means that not all children experience equally positive consequences of favourable relationship experiences in the context of their social development because of their biological aspects in form of their temperament. Our findings, thus, support previous theories in that there is not only one personality dimension (i.e., temperament or character), but the interaction of the two-factorial structure of personality and the parent–child interaction contribute to psychosocial development. This, in turn, emphasizes the importance of (re)building relationships and secure attachments by focusing on temperament and character dimensions. This may enhance the protective factor of caregiver–child interaction for highly burdened children and their (psychological) development. Notably, the small sample sizes in our study limited our ability to examine subgroups such as the relationship of temperament and character traits with attachment disorder types which needs further attention in future studies in the research field.

Our findings further showed associations of attachment disorder types with parental stress and child’s mental health problems. Our findings regarding mental health problems are in line with previous studies: first, the correlation in the meta-analyses by Groh et al. [7,40] between attachment and temperament with internalizing problems is smaller than the correlation with externalizing problems [41], and the heterogeneity for this effect size is mainly due to methodological differences and concept definitions between studies. Due to the small sample size and the cross-sectional nature of our study, further longitudinal research examining the causal relationship of emotional and behavioral problems with attachment disorders among larger samples is needed to obtain conclusive findings. Groh et al. [7], indeed, summarize that the strongest association between early attachment security appears to be with children’s later social interactions with peers, which once again highlights the relevance of attachment for individual’s healthy development. Second, our findings regarding parental stress is also in line with previous research [42,43,44,45]. In particular, the study of Bauch et al. [45] showed that the reduction of parental stress significantly reduced the occurrence of child neglect. Therefore, not only young people but also parents and other caregivers need to be addressed in the support and psychosocial help. From the perspective of attachment theory, due to the interaction between parent and child, caregivers should definitively be involved in psychosocial interventions including the assessment of their resources and burdens. Accordingly, caregivers should be provided with psychosocial support to deal with their own stress and strains as well as parent–child interactions to further promote their children’s healthy development.

### Strengths and Limitations

The findings of the present study have to be interpreted in the light of some limitations that could be addressed by future studies in this research field. The first limitation relates to the cross-sectional design of the study: The cross-sectional nature of our data limited our ability to perform further sub-group analyses (e.g., gender effects) and the interplay between the variables (e.g., mediation analysis). Thus, the bivariate correlations and relationship between our study variables do not imply any causality.

Second, although the current study had a sufficient total sample size, we were limited in conducting sub-group analyses for the three subgroups due to their small sample sizes. Our sample size was further reduced due to missing data. Although our attrition bias analyses did not reveal any significant group differences between the excluded participants vs. the included participants for our study variables (except of significant gender differences), an attrition bias of our sample cannot be completely ruled out.

Third, although we used one additional observational measure to assess attachment (GEV-B), we mainly used proxy reports (RPQ and CBCL) by primary caregivers. The quality of the study could have been increased by using a multi-perspective approach as well as in interaction and attachment behavior. Further assessments by teachers or experts could not be realized with the limited resources of this study. In addition, different types of caregivers were included in the present study. As such, caregivers either consisted of biological parents (i.e., community sample and clinical sample) or foster parents (i.e., foster care sample). This might have influenced the results because foster parents may be better trained to identify attachment disorders and problem behaviors of their foster children and are less stigmatized to report them than biological parents.

Fourth, in our study we had a uncommon high prevalence of AD in the CS (6.7%) and low prevalence of secure AR, while in epidemiologic studies the prevalence ranged from 0.9% to 1.4% for AD [24] and from 36.6% to 75.6% for AR [46]. This higher prevalence rate of AD and lower prevalence rate of AR should be discussed even when it is not statistically significant in our sample. One potential explanation for this non-significant deviation could be a selection effect because we outlined the topic attachment in the advertising for the study. Accordingly, parents who are sensitized for attachment topics or have already perceived an attachment problem in their children are probably higher motivated to participate, so that distortions may have occurred as a result of a selection bias in the CS of our study. Nevertheless, this subsample still meets the expected parameters of the general population regarding dimensional and categorical psychopathology (see Table 1).

Finally, the assessment of attachment representation based on a projective measure, which may introduce respondent and interpretation bias. Furthermore, to include as many children with attachment disorders as possible, we oversampled by recruiting children through mental health services and the foster care system in addition to children from the general population. Therefore, our study sample is heterogeneous, which might have influenced the present findings. At childhood (i.e., 6 to 11 years), boys tend to be more prevalent than girls in these professional settings. Therefore, we had significantly more boys in our sample. In addition, the foster children and children in child and adolescent psychiatric institutions were significantly older than the children in the general population who were recruited from nursery schools and primary schools.

Despite these limitations, our study has major strengths which makes its approach to the research questions posed valuable. First, we are one of the limited numbers of studies which included children who are currently underrepresented in the field of attachment research.

Second, in conceptualizing this study, we addressed calls for further research on the relationship between temperament and attachment [7,8,13]. To accomplish this, we expanded the content of the temperament concept and examined the relationship of temperament to both attachment concepts—namely attachment representations and attachment disorders.

Third and lastly, we recruited children from three different populations which enabled us to investigate group differences. Children in psychosocial settings are often difficult to include in such time-consuming research studies. Additionally, such studies tend to be designed for only one of the subsamples, which hampers the direct comparison of the results between children in the general population, child and adolescent psychiatric institutions and residential care. Our study made such comparisons possible.

## 5. Conclusions

From a clinical perspective, our preliminary findings regarding the relationship of temperament and character, parental stress, and mental health problems with attachment disorder among children indicate the importance of different concepts and manualized interventions for the prevention of corrective attachment experiences in early childhood (for an overview see NICE, 2015 [47]). There is also a need to support highly burdened families and their children at an early stage; thus, our findings indicate the importance of early identification and prevention of risk and protective factors for children’s psychosocial development within their parental homes. Not only person-related factors such as temperament and character, but also a pathological parent–child interaction should be taken into account for the indication and planning psychosocial services [18,48]. This requires more targeted low-threshold services in clinics, at pediatricians and in day-care centers for families who are affected.

From a scientific perspective, our findings suggest the complexity of the interplay between resilience-promoting personality traits and the attachment between parent–child for a healthy development. Furthermore, besides psychological assessment of these traits, future studies should investigate physiological and neurobiological markers to better understand the responses with temperament variables to attachment disorder types. Studies with larger samples are needed to examine subgroups regarding personality traits and attachment disorder types to disentangle the interplay of personality and attachment developments. To this end, it would initially appear to be advantageous to design future longitudinal studies in such a way that they examine the relationship between temperament and the developmental perspective of attachment representations and from the developmental psychopathological paradigm of attachment disorders. Consequently, more longitudinal studies with standardized assessment instruments in different and larger samples are needed to systematically examine the causal relationship and interplay between temperament and character traits, including their attachment and mental health, to further support healthy child development.

In the recognition of these empirical findings, without wanting to neglect the human complexity and thus many other factors, on the one hand, a sensitization for the emergence of (social) impairments and developmental pathology is possible in the context of psychoeducation for young people, their parents and professionals and, at the same time, interdisciplinary as well as interprofessional intervention knowledge and possibilities for action in the psychosocial fields of work. On the other hand, and based on a preventive salutogenetic approach, the potential protective effects of attachment and relationship formation for human development are to be communicated to young people, their parents and professionals with at least the same intensity, as well as made emotionally tangible. Based on this heuristic, basic and applied research should then be pursued to address previously unanswered questions and to open up new areas of interest.

## Figures and Tables

**Table 1 ijerph-19-15458-t001:** Sociodemographic and descriptive characteristics of the total sample and differences between sub-samples.

Variable	Total (*N* = 125)	Clinical Sample (*n* = 63)	Foster Care Sample (*n* = 32)	Control Sample (*n* = 30)	Test Statistic
Age in years (*M* [*SD*])	7.14 (1.38)	7.33 (1.38)	7.52 (1.42)	6.33 (1.03)	χ^2^(2) = 13.92, *p* < 0.001 ***
Gender (*n* [%])					χ^2^(2) = 11.20, *p =* 0.003 **
Female	49 (39.2)	16 (25.4)	19 (59.4)	14 (46.7)	
Male	76 (60.8)	47 (74.6)	13 (40.6)	16 (53.3)	
Nationality (Swiss)	66 (52.8)	22 (34.9)	26 (81.3)	18 (60)	χ^2^ (2) = 19.10, *p* < 0.001 **
Personality traits (*M* [*SD*])					
Novelty Seeking	52.82 (12.41)	56.94 (12.31)	51.94 (11.09)	45.1 (10.20)	χ^2^(2) = 20.76, *p =* 0.001 ***
Harm Avoidance	52.05 (10.77)	53.54 (10.19)	50.84 (11.29)	50.2 (11.29)	χ^2^(2) = 3.69, *p =* 0.16
Reward Dependence	44.82 (11.91)	40.63 (11.48)	50.31 (11.40)	47.77 (10.26)	χ^2^(2) = 15.55, *p =* 0.001 ***
Persistence	44.32 (11.54)	41.67 (10.74)	43.09 (12.25)	51.2 (9.82)	χ^2^(2) = 16.86, *p <* 0.001 ***
Self-directedness	43.74 (11.64)	40.14 (9.43)	44.06 (13.60)	50.93 (10.47	χ^2^(2) = 16.10, *p* < 0.001 ***
Cooperativeness	44.56 (12.89)	39.75 (11.29)	47.03 (13.16)	52.03 (11.73)	χ^2^(2) = 21.48, *p* < 0.001 ***
Self-transcendence	45.71 (10.06)	43.73 (10.77)	46.78 (9.52)	48.73 (8.28)	χ^2^(2) = 5.44, *p* = 0.07
Attachment disorder symptoms (*M* [*SD*])					
Total scale	4.65 (4.91)	6.05 (5.05)	4.75 (5.16)	1.6 (2.49)	χ^2^(2) = 24.70, *p* < 0.001 ***
Disinhibited subscale	2.35 (2.83)	2.90 (2.83)	2.66 (3.32)	0.87 (1.50)	χ^2^(2) = 24.70, *p* < 0.001 ***
Inhibited subscale	2.30 (2.98)	3.14 (3.49)	2.09 (2.36)	0.73 (1.36)	χ^2^(2) = 14.04, *p* < 0.001 ***
Attachment disorder (*n* [%])	33 (26.4)	19 (30.16)	12 (37.5)	2 (6.7)	χ^2^(2) = 8.50, *p* = 0.01 *
Attachment disorder type (*n* [%])					χ^2^(6) = 10.4, *p* = 0.08
None	92 (73.6)	44 (69.8)	20 (62.5)	28 (93.3)	
Inhibited	9 (7.2)	6 (9.5)	2 (6.3)	1 (3.3)	
Disinhibited	11 (8.8)	5 (7.9)	5 (15.6)	1 (3.3)	
Mixed-type	13 (10.4)	8 (12.7)	5 (15.6)	0 (0)	
Attachment relationship types					χ^2^(6) = 13.24, *p* = 0.04 *
Avoidant	22 (17.6)	12 (19.0)	6 (18.8)	4 (13.3)	
Secure	37 (29.6)	15 (23.8)	10 (31.3)	12 (40.0)	
Ambivalent	19 (15.2)	6 (9.5)	4 (12.5)	9 (30.0)	
Disorganized	47 (37.6)	30 (47.6)	12 (37.5)	5 (16.7)	
Parental stress (*M* [*SD*])	38.56 (9.21)	39.25 (9.68)	37.84 (8.52)	37.87 (9.10)	χ^2^(2) = 0.80, *p* = 0.67
Mental health problems (*M* [*SD*])					
Internalizing problems	58.96 (11.35)	63.60 (10.56)	56.81 (10.77)	51.5 (8.88)	χ^2^(2) = 25.25, *p <* 0.001 ***
Externalizing problems	61.68 (11.27)	66.11 (10.96)	60.31 (10.62)	53.83 (7.56)	χ^2^(2) = 26.96, *p* < 0.001 ***
Total problem behavior	62.21 (11.39)	67.46 (9.86)	60.41 (11.23)	53.1 (7.90)	χ^2^(2) = 33.10, *p* < 0.001 ***

*Note.* M = mean. SD = standard deviation. * *p* < 0.05, ** *p* < 0.01, *** *p* < 0.001.

**Table 2 ijerph-19-15458-t002:** Correlation matrix of all study variables.

	1.	2.	3.	4.	5.	6.	7.	8.	9.	10.	13.	14.	15.	16.	17.	18.	19.	20.	21.	22.
1. Age	1																			
2. Gender	0.14	1																		
3. Nationality	0.19	−0.17	1																	
4. Novelty seeking	−0.03	−0.19	0.10	1																
5. Reward dependence	−0.31	0.14	−0.15	−0.00	1															
6. Persistence	−0.28	−0.05	−0.21	−0.24	0.37 **	1														
7. Cooperativeness	−0.19	0.14	−0.16	−0.79 ***	0.30	0.44 ***	1													
8. Harm avoidance	0.15	0.17	−0.06	0.08	−0.04	−0.19	−0.17	1												
9. Self-directedness	−0.28	0.10	−0.12	−0.44 ***	0.33 *	0.60 ***	0.59 ***	−0.50 ***	1											
10. Self-transcendence	−0.06	0.19	−0.14	−0.03	0.33 *	0.38 **	0.11	0.08	0.28	1										
13. Internalizing problems	0.37 **	−0.13	0.18	0.43 ***	−0.34 *	−0.37 **	−0.55 ***	0.58 ***	−0.66 ***	−0.10	1									
14. Externalizing problems	0.20	−0.11	0.22	0.76 ***	−0.24	−0.32 *	−0.78 ***	0.09	−0.52 ***	−0.11	0.59 ***	1								
15. Total problem behavior	0.30	−0.19	0.20	0.70 ***	−0.32 *	−0.39 ***	−0.74 ***	0.37 **	−0.69 ***	−0.12	0.87 ***	0.87 **	1							
16. Parental stress	0.04	−0.01	0.04	0.34 *	−0.13	−0.25	−0.42 ***	0.14	−0.35 **	−0.00	0.31	0.41 ***	0.42 ***	1						
17. Attachment disorder symptoms (total scale)	0.12	−0.03	0.05	0.53 ***	−0.10	−0.27	−0.50 ***	0.13	−0.48 ***	−0.09	0.47 ***	0.59 ***	0.64 ***	0.38 **	1					
18. Attachment disorder symptoms (disinhibited subscale)	0.00	0.01	0.02	0.38 **	0.11	−0.26	−0.29	0.05	−0.34 *	−0.10	0.25	0.37 **	0.40 ***	0.27	0.84 ***	1				
19. Attachment disorder symptoms (inhibited subscale)	0.19	−0.06	0.05	0.52 ***	−0.27	−0.19	−0.55 ***	0.17	−0.46 ***	−0.06	0.55 ***	0.62 ***	0.68 ***	0.37 **	0.85 ***	0.43 ***	1			
20. Attachment disorder	0.17	0.08	−0.02	0.26	−0.07	−0.26	−0.31	0.01	−0.16	−0.08	0.25	0.28	0.29	0.23	0.44 ***	0.40 ***	0.34 *	1		
21. Attachment disorder type	0.16	0.07	0.02	0.30	−0.04	−0.29	−0.33 *	−0.03	−0.19	−0.09	0.25	0.28	0.31 *	0.26	0.56 ***	0.53 ***	0.44 ***	0.91 ***	1	
22. Attachment representation	−0.05	−0.16	0.17	0.15	−0.01	−0.11	−0.09	−0.06	−0.03	0.16	−0.03	0.09	0.04	−0.05	0.07	0.13	−0.02	0.09	0.08	1

* *p* < 0.05, ** *p* < 0.01, *** *p* < 0.001.

**Table 3 ijerph-19-15458-t003:** Descriptive characteristics for temperament and character, mental health problems, and parental stress with attachment disorder type.

	Attachment Disorder Type					
Personality Trait	None (*n* = 92)	Inhibited (*n* = 9)	Disinhibited (*n* = 11)	Mixed-Type (*n* = 13)	Test Statistic	Cohen’s *d*
Temperament (*M* [*SD*])						
Novelty Seeking	50.86 (12.28)	52.67 (8.87)	52.18 (10.78)	64.54 (10.46)	χ^2^(3) = 14.15, *p* = 0.002 **	0.63
Harm Avoidance	51.96 (10.79)	57.00 (10.79)	50.00 (11.18)	51.00 (10.44)	χ^2^(3) = 2.54, *p* = 0.47	0.00
Reward Dependence	45.32 (11.93)	40.33 (13.41)	45.91 (11.35)	43.54 (11.77)	χ^2^(3) = 1.22, *p* = 0.75	0.20
Persistence	46.10 (11.36)	43.00 (10.72)	40.36 (10.60)	36.00 (10.55)	χ^2^(3) = 9.46, *p* = 0.02 *	0.46
Character (*M* [*SD*])						
Self-directedness	44.85 (11.59)	42.00 (12.29)	44.82 (12.94)	36.15 (8.16)	χ^2^(3) = 6.89, *p* = 0.08	0.35
Cooperativeness	46.95 (12.87)	37.67 (8.87)	45.00 (11.36)	32.08 (10.78)	χ^2^(3) = 18.00, *p* < 0.001 ***	0.74
Self-transcendence	46.22 (10.56)	45.78 (7.93)	43.73 (9.05)	43.77 (9.01)	χ^2^(3) = 1.44, *p =* 0.70	0.20
Mental health problems (*M* [*SD*])						
Internalizing problems	57.89 (11.09)	65.89 (8.96)	56.82 (11.54)	67.69 (9.43)	χ^2^(3) = 13.37, *p =* 0.003 **	0.63
Externalizing problems	59.83 (11.26)	68.00 (8.09)	60.82 (10.34)	71.25 (7.99)	χ^2^(3) = 16.04, *p =* 0.001 **	0.70
Total problem behavior	60.27 (10.98)	67.78 (9.34)	60.45 (11.59)	73.54 (7.90)	χ^2^(3) = 18.10, *p <* 0.001 ***	0.13
Parental stress (*M* [*SD*])	37.33 (8.98)	41.44 (8.38)	37.45 (8.10)	46.31 (8.96)	χ^2^(3) = 10.64, *p* = 0.01 **	0.51

*Note.* M = mean. SD = standard deviation. * *p* < 0.05; ** *p* < 0.01; *** *p* < 0.001.

## Data Availability

The data presented in this study are available on request from the corresponding author.

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
