# Peer review of "The Relationship of Temperament and Character, Parental Stress, and Mental Health Problems with Attachment Disorders among Children"

_ijerph, 2022, doi:10.3390/ijerph192315458_

Round 1

Reviewer 1 Report

1- On page 1, 366-43 there is a brief presentation of the key concepts of the research. In my opinion the reader needs a more precise explanation of these concepts, especially of “character” that from classical psychology until today has been a very controversial concept and frequently abandoned because of its proximity to the concept of personality.

2- Some key concepts are missing, such as: “personality” although in (2, 60) the concept of Personality and its relationship with Attachment is mentioned, but  without substantiating or explaining that relationship. Cloninger's Theory should also be explicitly stated in the text and not taken for granted as it may result unclear to the reader.

3- The two previous points would be clearer if the article started with a precise definition of all the  basic elements of the research.

4- On page 3, 13-13 the sample is described, but the number of final subjects should be stated in the text. It  can be calculated from  the tables but it should also be indicated in the text

5- On pages 3 and 4   the tests used for the evaluation are mentioned, but without the psychometric characteristics of the adaptations used. These data should be added.

6- On page 4,16-17 under the headline “Personality traits” the instrument for the evaluation of Temperament and Character is presented. This cannot be admitted  unless the relations between these three concepts is previously justified in the introduction. In the description of the test, the concept of Character is explained,  but that should go much earlier in the paragraphs on Key concepts

7- On page 6, 21-22 only the obtained correlations consistent with the nomological framework are commented (table 2), while those inconsistent are not . For instance, the negative correlation between Persistence and Age, Internalized Problems and Turn off with Novelty seeking. These facts need an explanation in the Discussion.

8- In the limitations section, reference is made to the difficulty a reduced sample size represents, but this caution does not appear in the discussion or in the conclusions.

Author Response

Point-to-Point Response to Peer-Reviewer’s Comments

Reviewer #1:

“1:1- On page 1, 366-43 there is a brief presentation of the key concepts of the research. In my opinion the reader needs a more precise explanation of these concepts, especially of “character” that from classical psychology until today has been a very controversial concept and frequently abandoned because of its proximity to the concept of personality.

RESPONSE: We thank the reviewer for this important feedback and these comprehensive suggestions. Accordingly, we have implemented these recommendations in a fundamental didactic redesign and addition of content to the introduction. In this context, it has been important to us to provide an overview of the basic concepts so that we can continue to keep the reader well focused on the research questions. You will find the corresponding changes in the revised manuscript:

Lines 65-122:

“In personality psychology, many different but also overlapping attempts at defining human personality have developed since its origins due to different substantive emphases. The most relevant paradigms here are the psychoanalytic, behavioral/social learning, biological, evolutionary, neuroscientific and cognitive approaches. Consequently, there cannot be one definition of personality, as it varies according to the paradigm. However, a simple description of personality that summarizes these different paradigms makes it clear that personality is characterized by consistent patterns of behavior across time, space and situations that develop from the interaction between environment and individual in intrapersonal emotional, motivational and cognitive processes [1, 2].

Within personality theory, Robert C. Cloninger's psychobiological model of personality combines many of the paradigms listed above. Cloninger's concept of personality [3] makes the helpful conceptually operationalized distinction between temperament traits (predominantly genetic influences) and character traits (predominantly environmental influences) and considers the mutual influences of these traits in the process of personality development. Temperament consists of relatively consistent, basic and individual dispositions that underlie and modulate the expression of activity, reactivity, emotionality and sociability. Essential elements of temperament are present early in life, and these elements are strongly influenced by neurobiological factors. In Cloninger's model, temperament traits include novelty seeking, harm avoidance, reward dependence and persistence.  Novelty seeking represents the level of behavioral activation, harm avoidance represents the behavioral inhibition system, reward dependence represents the maintenance of behavior through social reinforcement, and the maintenance of behavior without external influences, i.e. through intrinsic motivation, is called persistence [4]. In a recent publication, Cloninger et al. [5] define Temperament as the disposition of a person to learn how to behave, react emotionally, and form attachments automatically by associative conditioning.

Character, the second basic personality dimension, includes self-concepts and individual differences in goals and values that influence decision-making, intentions and the meaning of what is experienced in life. Differences in character are said to be influenced by socio-cultural learning and to mature continuously in the life process. In Cloninger´s model of personality, character traits include self-directedness, cooperativeness, and self-transcendence. Self- directedness describes responsible and mature behavior and self-acceptance. The engagement with the social environment is depicted in cooperativeness, which is defined as helpful, tolerant and empathetic behavior. Self-transcendence, indicates the awareness of spiritual values, i.e. the extent of an individual's ability to recognize that other universal and transcendent values exist alongside the self and the social environment [4]. Impairments in the two character dimensions of self-directedness and cooperativeness are strong indicators of the presence of a personality disorder [3] which are closely related to attachment styles [6]. Cloninger's concept of personality considers postulates the interrelationship of continuous transactions between these four temperaments and character traits and three character traits in the process of personality development maturation [3].

There is still an unresolved question in developmental psychology about the relationship between attachment and personality or temperament and its developmental effects later in life [7, 8]. On the one hand, attachment has been described as an expression of temperament development [9]. Whereas, on the other hand, temperament development has been found to have little influence on attachment [10]. As a result, studies so far revealed contradictory findings, leading to ambiguous conclusions regarding the relationship between temperament, character and attachment [7, 8]. As a possible explanation for these unclear results, Groh et al [7] discussed the different operationalization of temperament. In addition, Bowlby [11] highlighted the complexity of parent-child interaction from an attachment perspective. Children have genetic dispositions that significantly influence their interaction with the environment, while biographical experiences of interaction with their parents substantially shape children's thoughts, feelings, and behaviors in close relationships. Concerning the relationship of attachment style and character, Chotai et al. [12] describe that secure attachment is positively correlated with cooperativeness and self-transcendence. Anxious/ambivalent attachment was correlated negatively with self-directedness.”

“2- Some key concepts are missing, such as: “personality” although in (2, 60) the concept of Personality and its relationship with Attachment is mentioned, but  without substantiating or explaining that relationship. Cloninger's Theory should also be explicitly stated in the text and not taken for granted as it may result unclear to the reader.”

 RESPONSE: We thank the reviewer for this important input. In accordance with your first comment, we have added content to the basic description of the personality concept via the presentation of the most relevant paradigms and described the Cloninger's concept in an overview manner. Due to the didactic and content-related short transitions, you will find the corresponding revisions made to your first reviewer's commentary.

“3- The two previous points would be clearer if the article started with a precise definition of all the  basic elements of the research.”

RESPONSE: As you already summarise yourself, we also estimate that by editing the first two reviewer comments to address your third reviewer comment. We thank you very much for these suggestions, so that the manuscript has received an even more stringent introduction to the topic and can thus better accompany the readership. Accordingly, we ask you to consider the revisions of your first commentary in the manuscript for this reviewer commentary.

“4- On page 3, 13-13 the sample is described, but the number of final subjects should be stated in the text. It can be calculated from  the tables but it should also be indicated in the text.”

RESPONSE: We thank the reviewer for this remark. We checked the section on page 3. The total number of final subjects for the present study is N = 125 which we added accordingly in the text. Also, we added the total number of subjects in the project (before excluding for missing data for our study) which is N = 152. We also emphasized the final sample by adding the ‘a total of’ to it. Both information can be found in the revised manuscript and reads as follows:

Lines 166-169: “The aim of the study was to assess attachment representations and disorders among a total of 152 children from the general population (control sample; n = 34, mean age = 7.52 years), foster care (foster care sample; n = 32, mean age = 7.52 years), and clinical settings (clinical sample; n = 86, mean age = 7.24 years).”

Lines 195-197: The final sample of the present study consisted of a total of 125 children (age range = 4–10 years) including children from the general population (n = 30), foster care (n = 32), and child and adolescent psychiatric settings (n = 63).

“5- On pages 3 and 4   the tests used for the evaluation are mentioned, but without the psychometric characteristics of the adaptations used. These data should be added.”

RESPONSE: We thank the reviewer for this remark. We added information regarding the psychometric properties of all questionnaires based on previous studies in our “2.3 Measures”-section. For each measure, we added the information regarding psychometric properties of the adapted version which reads as following in the revised manuscript:

Lines 224-225: “The K-DIPS has good validity and reliability for axis-I disorders (parent version, kappa = 0.88 to 0.95; (23)).”

Lines 235-238:The German version of the RPQ shows high internal consistencies for the RPQ total score (Cronbach’s a= 0.81) as well as for the disinhibited (Cronbach’s α= 0.86) and inhibited (Cronbach’s a=0.74) attachment behavior subscales (27).”

Lines 243-245:Previous studies showed good 168 interrater reliability with regard to their assessments of the attachment security value (28).”

Lines 252-254:The questionnaire has shown good psychometric properties in a German normative and clinical sample (30).”

Lines 260-261: “The German version of the PSS showed a good internal consistency and construct validity (33).”

Lines 268-269:The internal consistency and construct validity of the German version of the CBCL was found to be good in previous studies (35).”

“6- On page 4,16-17 under the headline “Personality traits” the instrument for the evaluation of Temperament and Character is presented. This cannot be admitted  unless the relations between these three concepts is previously justified in the introduction. In the description of the test, the concept of Character is explained,  but that should go much earlier in the paragraphs on Key concepts.”

RESPONSE: We thank the reviewer for this remark. According to the first three reviewer comments in combination with this precinct note, we have already placed the description of these three concepts to the introduction, so we would like to ask you to consider this edit. Furthermore, we have decided to replace the heading "Personality traits" with the heading "Temperament and character traits" in order to comply with the operationalization for personality, but especially for temperament, that is required in the scientific discourse. Following Cloninger's model of personality, we thus follow the description in temperament and character traits. With the renaming of the heading, this becomes clearer and more stringent in the manuscript. The corresponding change in the manuscript can be found here:

Line 247:Temperament and character traits.”

“7- On page 6, 21-22 only the obtained correlations consistent with the nomological framework are commented (table 2), while those inconsistent are not. For instance, the negative correlation between Persistence and Age, Internalized Problems and Turn off with Novelty seeking. These facts need an explanation in the Discussion.”

RESPONSE: We understand the reviewer’s attentive input. First, we would like to add that we reported only the findings regarding our main hypothesis – namely the relationship of temperament and character and attachment disorders – due to the limited number of words count and pages of the journal. Also, we took this as an occasion to re-check our data and correlation matrix and we indeed had to correct some of our correlations/numbers in the Table 2. As saying so, for example, the correlation reported in the table for age and persistence was wrong, and not significant actually (correct r = -0.28, not significant). We thus updated and corrected our Table 2 as well as the results section in the text accordingly.

We understand the reviewer's interest in explanation the correlation between the individual variables, but in our view, this loses focus on the research question about the relationship between temperament, character, and attachment, which is our main concern with this manuscript. Moreover, for the quality of analysis needed at this level, profiling would be required to map the complex interactions, which would be a research question in its own right and would require the scope of another separate article. In addition, the majority of potential inconsistent results emerged as errors during the data review, so we will not go into the explanations of the correlations between the individual variables. While we follow all the other advices of the reviewer, we ask for your understanding for this.

“8- In the limitations section, reference is made to the difficulty a reduced sample size represents, but this caution does not appear in the discussion or in the conclusions”

RESPONSE: We would like to thank the reviewer for this remark. We have addressed the issue regarding our reduced sample size in our discussion as well as conclusion sections and included it for the interpretation of our findings. The added information in the discussion and conclusion sections reads as follows in the revised version of our manuscript:

Lines 389-392: “Notably, the small sample sizes in our study limited our ability to examine subgroups such as the relationship of temperament and character traits with attachment disorder types which needs further attention in future studies in the research field.”

Lines 399-402: “Due to the small sample size and the cross-sectional nature of our study, further longitudinal research examining the causal relationship of emotional and behavioral problems with attachment disorders among larger samples is needed to get conclusive findings.”

Lines 492-495: “Studies with larger samples are needed to examine subgroups regarding personality traits and attachment disorder types to disentangle the interplay of personality and attachment developments.”

 Lines 498-502: “Consequently, more longitudinal studies with standardized assessment instruments in different and larger samples are needed to systematically examine the causal relation-ship and interplay between temperament and character traits including their attachment and mental health to further support a healthy child development.”

Reviewer 2 Report

Dear authors,

In Table 2, the correlation between Novelty seeking and Self-directedness is probably .50, not .05, if it is statistically significance at p < 0.001, right? Please correct it.

Author Response

Point-to-Point Response to Peer-Reviewer’s Comments

Reviewer #2:

  1. “Dear authors, In Table 2, the correlation between Novelty seeking and Self-directedness is probably .50, not .05, if it is statistically significance at p < 0.001, right? Please correct it.”

RESPONSE: We thank the reviewer for this comment, and we fully agree. We checked this issue in our data analysis and we have noticed that we made a mistake when creating the table regarding the number figures of that correlation coefficient. Therefore, we are very thankful for your attentive remark of reviewer 2 for noticing this mistake. We also noticed that the number figures for the correlation coefficient was only wrong in the Table 2 but correct in the text. Therefore, we corrected that number figure in the revised version of Table 2 and changed it from r = -0.05*** to r = -0.46***.

Reviewer 3 Report

The authors used "The relationship of temperament and character, parental stress, and mental health problems with attachment disorders among children in middle childhood" as the research topic, which affirms the research value and importance of this topic.

L97: "The aim of the present study was to assess attachment representations and disorders among children in middle childhood from the general population (control sample). The general population (control sample), foster care (foster care sample), and clinical settings (clinical sample)" from the perspective of the study, it is considered that the authors collected the study data in different ways. However, we ask the authors to add information about the number of samples and the main age groups in each field.

L121:

(1) The study sample consisted of 125 children, aged between 4 and 10 years old, compared to the name "middle childhood" in this study. (2) The authors are requested to add the age range of "middle childhood."

(2) Could the authors please add why the age group of 4 to 10 years old was finally adopted as the main study group? Is it representative of the study? Please also indicate the number of people in each age range of the study group.

112:The study design included a combination of questionnaires and face-to-face.

L112: The study design included a combination of questionnaires and face-to-face assessments, and there is a significant cognitive and comprehension gap between children aged 4-10 years old. Including the implementation time? Please also add details on how collected the data for this study.

L261: In the study's conclusion, the authors present the post-statistical data, and most of them are consistent with Groh's study. In the hope that this paper can make more academic contributions, please ask the authors to present more specific findings on the research questions in L90, and if they can supplement the narrative nature of the research design (L112-L116), the conclusion of this study will be more valuable.

Author Response

Point-to-Point Response to Peer-Reviewer’s Comments

 Reviewer #3:

“The authors used "The relationship of temperament and character, parental stress, and mental health problems with attachment disorders among children in middle childhood" as the research topic, which affirms the research value and importance of this topic.

  1. L97: "The aim of the present study was to assess attachment representations and disorders among children in middle childhood from the general population (control sample). The general population (control sample), foster care (foster care sample), and clinical settings (clinical sample)" from the perspective of the study, it is considered that the authors collected the study data in different ways. However, we ask the authors to add information about the number of samples and the main age groups in each field.

RESPONSE: We thank the reviewer for this attentive comment, and we fully agree. We provided the information of the sample sizes for each subgroup in the methods section. Furthermore, we calculated the mean age of all subgroups separately and added that information in the text. The section with the added information in the revised manuscript now reads as follows:

Lines 166-169: “The aim of the study was to assess attachment representations and disorders among a total of 152 children from the general population (control sample; n = 34, mean age = 7.52 years), foster care (foster care sample; n = 32, mean age = 7.52 years), and clinical settings (clinical sample; n = 86, mean age = 7.24 years).”

L121:

(1) The study sample consisted of 125 children, aged between 4 and 10 years old, compared to the name "middle childhood" in this study. (2) The authors are requested to add the age range of "middle childhood."

RESPONSE: We thank the reviewer for the valid comment. Because we included both early and middle childhood developmental stages in our study due to the age range of 4 to 10 years, we will remove the addition of middle childhood from the entire manuscript without replacement. Most clearly, and exemplary of this sweeping change is the adjusted title of the manuscript:

Lines 2-4: “The relationship of temperament and character, parental stress, and mental health problems with attachment disorders among children

(2) Could the authors please add why the age group of 4 to 10 years old was finally adopted as the main study group? Is it representative of the study? Please also indicate the number of people in each age range of the study group.

RESPONSE: We thank the reviewer for the reasonable inquiry. As you can already see in the response to your first comment, the main study consisted of 152 children, with a mean age of 7.52 and 7.24 years, depending on the subgroup. The 125 of the 152 children included in the analysis here correspond to this age range and are therefore representative of the main study. The addition of the number of children per age group of the subgroups requested by the reviewer was already implemented in the response to the first reviewer comment.

112: The study design included a combination of questionnaires and face-to-face.

L112: The study design included a combination of questionnaires and face-to-face assessments, and there is a significant cognitive and comprehension gap between children aged 4-10 years old. Including the implementation time? Please also add details on how collected the data for this study.

RESPONSE: We thank the reviewer for the clarifying inquiry. The study was indeed conducted in a combination of questionnaires and face-to-face. However, as you rightly point out with your reference to the cognitive and comprehension gap between children aged 4-10 years old, it requires clarifying specification. Due to the developmental age of the children, they only participated in the face-to-face assessment, German attachment story completion procedure (GASCP), and did not fill out any questionnaires in order not to overtax the children. The GASCP was explicitly developed for this developmental age of the children. The child explanation of the procedure, verbally obtaining the child's consent for implementation, introduction, implementation, and completion of the GASCP requires an average of 60minutes and was fully videotaped.

In contrast, the children's caregivers participated in the structured interview for axis-I disorders in parallel and received the necessary questionnaires in advance for processing. The structured interview with the K-Dips was directly written down in the corresponding coding system and the questionnaires were computerized for analysis. This general and specific helpful information can be found supplemented in the manuscript at the following lines:

Lines 177-184:The children only participated in the face-to-face assessment and did not fill out any questionnaires. The child explanation of the procedure, verbally obtaining the child's consent for implementation, introduction, implementation, and completion of the procedure requires an average of 60minutes and was fully videotaped. In contrast, the children's caregivers participated in the structured interview for axis-I disorders in parallel and received the necessary questionnaires in advance for processing. The structured interview with the K-Dips was directly written down in the corresponding coding system and the questionnaires were computerized for analysis.”

L261: In the study's conclusion, the authors present the post-statistical data, and most of them are consistent with Groh's study. In the hope that this paper can make more academic contributions, please ask the authors to present more specific findings on the research questions in L90, and if they can supplement the narrative nature of the research design (L112-L116), the conclusion of this study will be more valuable.

RESPONSE: We would like to thank the reviewer for the opportunity to elaborate on our larger research interest behind the presented study. Based on the brief presentation of the empirical findings in the introduction, we believe it is clear what protective as well as pathogenetic effect attachment can have on human development. In the recognition of this empirical findings without wanting to neglect the human complexity and thus many other factors, on the one hand a sensitization for the emergence of (social) impairments and developmental pathology is possible in the context of psychoeducation for young people, their parents and professionals and at the same time interdisciplinary as well as interprofessional intervention knowledge and possibilities for action in the psychosocial fields of work. On the other hand, and based on a preventive salutogenetic approach, the potential protective effects of attachment and relationship formation for human development are to be communicated to young people, their parents and professionals with at least the same intensity, as well as made emotionally tangible. Based on this heuristic, basic and applied research should then be pursued to address previously unanswered questions and to open up new areas of interest. We are happy to contribute these thoughts and ideas to the final discussion and would like to thank the reviewer for this opportunity:

Lines 503-513: “In the joint consideration of these two perspectives, there are main topics to be consider and deal with for the general population, the psychosocial settings of help as well as further research performance. Without wanting to neglect human complexity and thus many other factors, on the one hand a sensitization for the emergence of (social) impairments and developmental pathology is possible within the framework of psychoeducation for (young) people, their parents and professionals and at the same time interdisciplinary as well as interprofessional intervention knowledge and possibilities for action in the psychosocial settings of help. On the other hand, on the basis of a preventive-salutogenetic approach, the potentially protective effects of attachment and relationship for human development should be conveyed to young people, their parents and professionals with at least the same intensity and made emotionally tangible. Basic and applied research should then be conducted on the basis of these heuristics to clarify previously unanswered questions and to open up new areas of interest.”